# ExT5: Towards Extreme Multi-Task Scaling for Transfer Learning

**Vamsi Aribandi**[*†], **Yi Tay**[†], **Tal Schuster, Jinfeng Rao, Huaixiu Steven Zheng,**
**Sanket Vaibhav Mehta, Honglei Zhuang, Vinh Q. Tran, Dara Bahri, Jianmo Ni,**
**Jai Gupta, Kai Hui, Sebastian Ruder**♣**, Donald Metzler**
Google Research, ♣DeepMind
{aribandi, yitay}@google.com

## ABSTRACT

Despite the recent success of multi-task learning and transfer learning for natural language processing (NLP), few works have systematically studied the effect of scaling up the number of tasks during pre-training. Towards this goal, this paper introduces ExMix (**Ex**treme **Mix**ture): a massive collection of 107 supervised NLP tasks across diverse domains and task-families. Using ExMix, we study the effect of multi-task pre-training at the largest scale to date, and analyze co-training transfer amongst common families of tasks. Through this analysis, we show that manually curating an ideal set of tasks for multi-task pre-training is not straightforward, and that multi-task scaling can vastly improve models on its own. Finally, we propose ExT5: a model pre-trained using a multi-task objective of self-supervised span denoising and supervised ExMix. Via extensive experiments, we show that ExT5 outperforms strong T5 baselines on SuperGLUE, GEM, Rainbow, Closed-Book QA tasks, and several tasks outside of ExMix. ExT5 also significantly improves sample efficiency while pre-training.

## 1 INTRODUCTION

Transfer learning (Schmidhuber, 1987; Pratt et al., 1991; Caruana et al., 1995) has been the cornerstone of recent progress in natural language processing (Ruder et al., 2019; Devlin et al., 2019; Raffel et al., 2020). While self-supervised pre-training has been shown to be highly effective at exploiting large amounts of unlabeled data without relying on human annotation, there is still much to explore regarding transfer learning in a multi-task co-training setup.

Prior seminal works like T5 (Raffel et al., 2020) and MT-DNN (Liu et al., 2019a) have demonstrated a degree of promise in the paradigm of multi-task co-training (Caruana, 1997). However, the challenge of catastrophic forgetting remains. Tasks often have to be carefully selected in order to demonstrate positive affinity with regards to downstream transfer. In many cases, it is not unreasonable to expect negative transfer (Rosenstein et al., 2005; Vu et al., 2020). This makes the process of empirically curating a set of tasks to include in a transfer learning setup both computationally prohibitive and specific to downstream tasks.

While standard pre-training typically employs a variant of the self-supervised language modeling objective (Raffel et al., 2020), certain types of skills such as commonsense knowledge are only acquired at a slow rate even using massive amounts of unlabeled data (Zhang et al., 2021). As ever larger models are trained, the development of much more sample-efficient pre-training settings becomes thus more important, and could be addressed via multi-task learning.

For the first time, we explore and propose *Extreme Multi-task Scaling* — a new paradigm for multi-task pre-training. Compared to the largest prior work (Aghajanyan et al., 2021), our study doubles the number of tasks and focuses on multi-task pre-training rather than fine-tuning, which enables a direct comparison to standard pre-training. Our proposal is based on the insight that despite negative transfer being common during fine-tuning, a massive and diverse collection of pre-training tasks is generally preferable to an expensive search for the best combination of pre-training tasks.

---

[*]Google AI Resident. [†]Equal contribution. Sebastian is now at Google Research. Sanket returned to CMU.

To this end, we introduce ExMix: a massive collection of 107 supervised NLP tasks to be included in a multi-task pre-training setup. We process all tasks in an encoder-decoder friendly format to readily support the sharing of all parameters across all tasks. We postulate that an *ensembling* effect across as many tasks, distributions and domains as possible results in a consistently net-positive outcome. This echoes early multi-task learning results (Caruana, 1997; Baxter, 2000) indicating that a bias that is learned on sufficiently many tasks is likely to generalize to unseen tasks drawn from the same environment. Moreover, our experiments verify that our ExMix mixture outperforms a best-effort mixture of manually curated tasks.

Finally, we propose ExT5: a T5 model (Raffel et al., 2020) pre-trained on a mixture of supervised ExMix and self-supervised C4 span denoising. ExT5 outperforms state-of-the-art T5 models on well-established benchmarks such as SuperGLUE (Wang et al., 2019a), GEM (Gehrmann et al., 2021), and Rainbow (Lourie et al., 2021); as well as Closed-Book QA (Roberts et al., 2020) tasks. Notably, our experimental findings also suggest that including ExMix may reduce the number of pre-training steps required to achieve strong performance, bringing about substantial sample efficiency benefits.

To summarize, this paper contributes the following:

– We propose ExMix (§2): a collection of 107 supervised NLP tasks for *Extreme Multi-task Scaling*, formatted for encoder-decoder training. ExMix has approximately twice as many tasks as the largest prior study to date (Aghajanyan et al., 2021), totaling 18M labeled examples across diverse task families.

– Given this large collection of tasks, we conduct rigorous empirical studies evaluating transfer between common task families (§2.1). Our experiments show that curating a pre-training mixture based on fine-tuning transfer is not straightforward (§2.2). Hence, efficiently searching for the best subset of tasks to include in a multi-task pre-training setup is challenging and prohibitive.

– Using ExMix, we pre-train a model alongside the C4 span-denoising objective introduced by Raffel et al. (2020), resulting in a new pre-trained model which we call ExT5 (§3). ExT5 outperforms state-of-the-art T5 on well-established benchmarks such as SuperGLUE, GEM, Rainbow, Closed Book Question Answering, and several other tasks that are outside of ExMix (§3.2), while also being more sample-efficient (§2.6).

## 2 THE EXMIX TASK COLLECTION

To explore the Extreme Task Scaling paradigm, we introduce ExMix (**Ex**treme **Mix**ture), a collection of 107 diverse English NLP tasks totaling 18M examples. Following Raffel et al. (2020), we format all tasks as text-to-text examples to readily allow for multi-task training. This unified format also enables simple implementations without the need for task-specific heads/losses, loss scaling, or explicit gradient accumulation for heterogenous batches as in prior works (Liu et al., 2019a; Aghajanyan et al., 2021). When selecting examples from ExMix, ex-

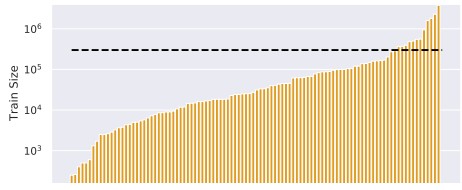

Figure 1: ExMix task sizes in log scale. The dashed line is the $3 \times 10^5$ sampling rate cap.

amples from each dataset are sampled proportionate to the individual dataset's size, with each dataset's sampling rate capped at $3 \times 10^5$ maximum effective examples to ensure a balance between large and small datasets. We refer readers to Appendix A for a comprehensive breakdown of ExMix. Additionally, we discuss future multilingual variants of ExMix and ExT5 in §5.

### 2.1 TRANSFER RELATIONS BETWEEN EXMIX TASKS

As discussed in §1, our goal is to pre-train a model on ExMix to improve downstream performance. One natural question to ask is *which tasks have a negative impact on downstream performance?* Specifically, is there a subset of ExMix that leads to better representations when used for multi-task pre-training? Obviously, testing all possible $2^{|ExMix|}$ combinations is impractical since the

pre-training process is expensive to run. Instead, we experiment with the less expensive *co-training* procedure (i.e., multi-task fine-tuning), using representative subsets of similar tasks. Later, in §2.2, we explore whether these results can be used to inform the task selection for multi-task pre-training.

To study transfer amongst task-families in EXMIX, we construct subsets (Table 1) of 3 tasks each that are partitioned along their task family. Using these subsets, we evaluate transfer among task families in a multi-task learning (co-training) setup. While other types of tasks are available in EXMIX, we did not include them because they were not diverse enough to be representative of a task family, and would scale the number of models needing to be trained at a quadratic rate.

**Experimental Setting** We fine-tune a model on each pair of task families (i.e., 6 datasets at a time). To ensure a fair balance of tasks, we sample tasks proportional within their family, but uniformly between task families. For example, while evaluating how classification tasks and NLI tasks transfer amongst each other, the sampling ratio of MNLI:ANLI will be proportional (approximately 2.4:1), but the overall ratio of NLI examples to classification examples will be 1:1. For reference, we also train a model on each individual task family using proportional example mixing (Sanh et al., 2019).

| Task Family | Datasets |
|---|---|
| Summarization | CNN/DailyMail
XSum
Wiki Lingua |
| Dialogue | Schema-guided dialogue
Wizard-of-Wikipedia
Dialoglue-TOP |
| NLI | ANLI
MNLI
$\alpha$NLI |
| Classification | IMDb reviews
GoEmotions
Civil Comments |
| Semantic Parsing | ATIS to FunQL
GEO to FunQL
COGS |
| Commonsense | PhysicaliQA
SocialiQA
WinoGrande |
| Closed-Book QA | Natural Questions
Trivia QA
Hotpot QA |
| Reading Comprehension | SQuAD
BoolQ
TweetQA |

Table 1: Representative datasets used for task-family transfer learning experiments (§2.1).

In total, this results in $F + \binom{F}{2}$ models trained, where $F$ is the number of task families. Our experiments use $F = 8$ as shown in Table 1, resulting in 34 models trained in total. Each model is fine-tuned on top of the released T5.1.1$_{\text{BASE}}$ checkpoint for 200k steps using a batch size of 128 and a constant learning rate of $10^{-3}$.

**Observations** We summarize our results in Table 2. We observe that although there exist particular task-family pairs that show positive transfer (e.g., co-training with NLI helps most other tasks), negative transfer is more common when training across task families compared to intra-family training. 21 out of the 56 inter-family relationships perform worse than intra-family models with the same data budget, which grows to 38 out of 56 for a fixed compute-budget. While the abundance of negative transfer among diverse task families is an expected result, interesting trends manifest in the individual relationships. For example, summarization tasks generally seem to hurt performance on most other task families; and CBQA tasks are highly sensitive to multi-task fine-tuning.

We also report correlations for intra-family datasets in Figure 2 using the same models as in Table 2. In most cases, we see positive correlations between datasets in the same family. In a few cases, however, we observe an opposite trend. For example, fine-tuned models that performed better on the GEM schema-guided dialog dataset achieved lower scores on KILT Wizard-of-Wikipedia.

This initial exploratory analysis highlights both the potential of EXMIX as a tool to systematically study task relationships, as well as the potential challenges in leveraging multi-task learning naively on top of pre-trained representations.

## 2.2 CAN FINE-TUNING TASK RELATIONSHIPS HELP CURATE A PRE-TRAINING MIXTURE?

Our observations in §2.1 showed that multi-task co-training on top of existing pre-trained checkpoints is not straightforward, and often results in negative transfer. However, the uncovered task relationships might help efficiently search for an ideal subset of EXMIX for multi-task pre-training. To this end, we select a set of the most promising task families to be included in a multi-task pre-training setup, ranking task families by the average relative improvement they provide to other target families (the last column in Table 2). Specifically, we include NLI, commonsense, classification, and closed-book QA tasks from EXMIX to form a mixture of 48 tasks to include in a multi-task

|  | SUM | DLG | NLI | CLS | SEM | CMNS | CBQA | RC | $\Delta_{\text{AVG}}$ |
|---|---|---|---|---|---|---|---|---|---|
| SUM | 27.89 / 29.36 | 37.81 | 60.45 | 77.10 | 78.25 | 61.92 | 7.84 | 65.37 | -6.9% |
| DLG | 29.05 | 38.56 / 39.76 | 63.62 | 77.10 | 75.55 | 64.05 | 13.39 | 64.75 | +0.1% |
| NLI | 28.61 | 40.60 | 64.91 / 67.23 | 77.29 | 77.72 | 67.60 | 15.24 | 66.40 | +4.3% |
| CLS | 29.52 | 40.16 | 66.69 | 77.14 / 77.47 | 76.05 | 65.29 | 12.93 | 65.20 | +1.4% |
| SEM | 29.30 | 38.86 | 62.46 | 76.83 | 72.09 / 72.79 | 57.84 | 12.44 | 63.52 | -2.5% |
| CMNS | 29.28 | 39.27 | 65.08 | 77.05 | 76.29 | 68.24 / 68.35 | 16.48 | 66.01 | +4.7% |
| CBQA | 29.75 | 39.29 | 64.96 | 77.66 | 75.21 | 66.84 | 14.68 / 19.98 | 66.37 | +1.2% |
| RC | 29.45 | 38.12 | 63.70 | 77.14 | 76.98 | 66.62 | 10.26 | 62.94 / 65.60 | -2.4% |
| $\text{AVG}_{\backslash \text{diag}}$ | 29.28 | 39.16 | 63.77 | 77.17 | 76.43 | 64.31 | 12.65 | 65.37 |  |

Table 2: Co-training transfer among task families. The entry at (row $i$, column $j$) indicates average performance on family $j$ using a model co-trained on families $i$ and $j$. For intra-family models (diagonal cells) we report results upto 100k steps (consistent data-budget) and 200k steps (consistent compute-budget). Averages are calculated excluding the intra-family models (i.e. the diagonal cells). The last column denotes the average gain that a source family provides to other task families.

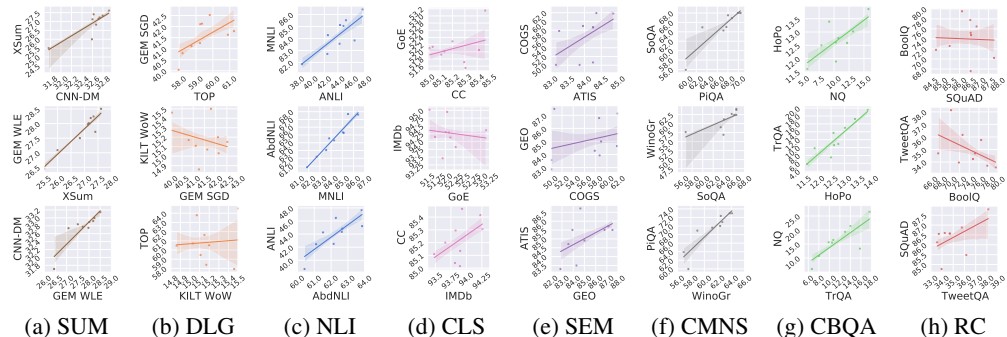

(a) SUM  (b) DLG  (c) NLI  (d) CLS  (e) SEM  (f) CMNS  (g) CBQA  (h) RC

Figure 2: Within-family correlations for each dataset in a task family, using models from Table 2. Performance on datasets from some task families are highly correlated (e.g., NLI) whereas other task families have more erratic results across their datasets (e.g., Dialogue)

pre-training setup. We then fine-tune the resulting model on SuperGLUE, comparing it to T5.1.1 and EXT5 in Table 3.

While this model narrowly outperformed T5.1.1, it did not yield better results than including all of EXMIX in the multi-task pre-training mixture, as we report in §3.2. Moreover, it did not outperform a random selection of 55 tasks on average, as we report in our task scaling experiments (§2.5).

We conclude that the **negative transfer during multi-task fine-tuning does not necessarily inhibit pre-training**. While we cannot directly conclude that an ideal subset of EXMIX does not exist to be mixed with self-supervised pre-training for specific downstream tasks, our experiments show that randomly including more diverse pre-training tasks generally improves downstream performance. It must also be noted that the end-goal is to find a mixture that leads to a *general* pre-trained model that can be used for a large variety of downstream tasks, and that a setup to find a pre-training mixture tailored for SuperGLUE would be different.

| Mixture | # Tasks | SuperGLUE |
|---|---|---|
| Vanilla | 0 | 76.1 |
| Best-effort | 48 | 76.4 |
| Random-55 (§2.5) | 55 | 77.0 |
| EXMIX (§3.2) | 107 | **79.9** |

Table 3: A best-effort mixture from fine-tuning transfer results does not beat increasing the number of tasks.

## 2.3 MULTI-TASK PRE-TRAINING VS PRE-FINETUNING

Instead of pre-training on ExMIX, another way to leverage multi-task learning is as an intermediate step between pre-training and fine-tuning. This is referred to as *pre-finetuning* by Aghajanyan et al. (2021). We conduct controlled experiments to compare pre-training with pre-finetuning. We begin with a standard T5 base checkpoint and pre-finetune it with ExMIX. After this phase, we fine-tune on SuperGLUE.

Table 4 compares pre-finetuning and our proposed multi-task pre-training. We also report the total compute (in total number of tokens processed) by the model in both schemes. The results show that multi-task pre-training is significantly superior to pre-finetuning. A potential hypothesis is that multi-task pre-training narrows the gap between pre-training and finetuning data distributions, as the pre-training stage more closely resembles fine-tuning. Conversely, segregating pre-training and pre-finetuning into two different stages may induce catastrophic forgetting of the pre-training task. Hence, in ExT5, we opt for multi-task pre-training over pre-finetuning.

| Method | Compute | SuperGLUE |
|---|---|---|
| Vanilla | 1.2M | 76.1 |
| Pre-finetuning (200k) | 1.4M | 78.1 |
| Multi-task Pre-training | 1.2M | **79.9** |

Table 4: Comparison of Pre-finetuning and Multi-task Pre-training on ExMIX.

## 2.4 HOW MUCH LABELED DATA SHOULD BE MIXED?

In this section, we explore how the ratio of C4 to ExMIX examples during massive multi-task pre-training affects performance. As mentioned later in §3, this is controlled by a hyperparameter $R$, where a pre-training batch will have approximately $R$ times as many C4 examples compared to ExMIX. From our results in Figure 3, we find that despite ExMIX improving downstream performance when mixed with self-supervised C4 pre-training at many rates, a model trained with $R = 0$ suffers greatly in comparison. This result is significant, as it shows that while ExMIX improves the pre-training process, self-supervised training over a large unstructured corpus is still crucial.

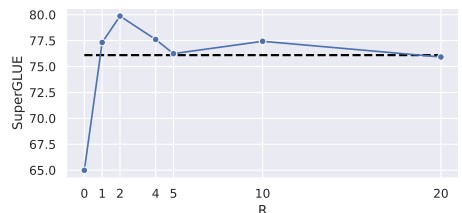

Figure 3: How the ratio of C4 span denoising examples to ExMIX affects SuperGLUE results on ExT5$_{\text{BASE}}$. The dashed line is performance without using ExMIX ($R \to \infty$)

## 2.5 DOES ADDING MORE TASKS HELP? TASK SCALING EXPERIMENTS

In this section, we explore how model performance changes as the number of tasks included in a massive multi-task pre-training setup is scaled up. We choose random sets of 30, 55, and 80 tasks (each a superset of the last), pre-train a BASE-sized model for 524k steps, and fine-tune them on SuperGLUE. We train our models with batch sizes of 128 and 512 and $R = 2$ (the ratio of C4 to ExMIX examples) as this configuration worked best for our BASE-sized models (§2.4). We repeat this over three random seeds (for random subset selection), and report average scores in Figure 4.

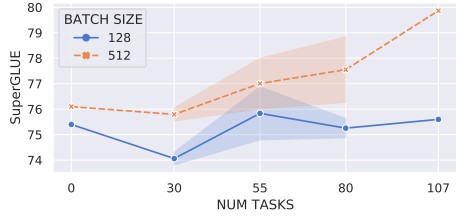

Figure 4: Scaling the number of tasks during multi-task pre-training generally helps. The shaded area portrays standard deviation across three random subset selections.

Overall, with large batches, we can see that increasing the number of tasks being mixed generally helps downstream performance. This reinforces our intuition that task scaling indeed helps. With small batches, there is less of an upward trend, signifying that large batches are essential for a large number of tasks. This is intuitive, given that multi-task learning may cause gradients to be noisy (Yu et al., 2020). Another explanation as to why this happens is that large-batch training can offer benefits even for single-task models (Smith et al., 2018) — a trend formalized by McCandlish et al. (2018).

## 2.6 Improving Sample Efficiency with ExMix

We hypothesize that extreme multi-task scaling also improves the sample efficiency of pre-training. To test this, we exclude SuperGLUE from ExMix, pre-train a large model for 200k steps, and fine-tune it on SuperGLUE at several intervals during early pre-training stages. We find that ExMix pre-training is significantly more sample-efficient than vanilla self-supervised pre-training. Note that at only 20k pre-training steps, our ExT5 model already achieves 75.8 SuperGLUE score, which outperforms a **fully pre-trained** BERT large model by about $+4\%$ (Wang et al., 2019a).

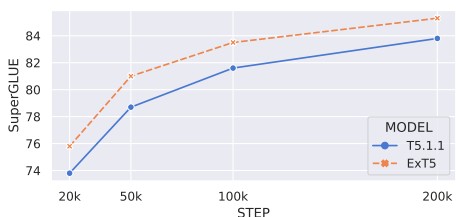

Figure 5: SuperGLUE score of ExT5$_{\text{LARGE}}$ vs T5$_{\text{LARGE}}$ as a function of number of pre-training steps.

## 3 The ExT5 Model

To overcome the challenges of multi-task co-training at scale, i.e. negative transfer and catastrophic forgetting explored in §2.1, the rest of this paper revisits the multi-task pre-training paradigm introduced by Raffel et al. (2020) via extreme multi-task scaling. This section introduces ExT5: a pre-trained sequence-to-sequence Transformer encoder-decoder model (Vaswani et al., 2017) based on the popular T5 framework.

### 3.1 Training ExT5

**Pre-training**   We pre-train on a mixture of C4 and ExMix (§2), and combine them with a hyper-parameter $R$ that is the ratio at which C4 examples are sampled with respect to ExMix examples. The C4 objective we use is the same as that used by Raffel et al. (2020), and every task optimizes the standard sequence-to-sequence cross-entropy loss. We pre-train ExT5 on the same number of steps as T5, and ExT5 sees an identical number of tokens to the released T5 models. Concretely, we pre-train our models for 1M total steps with a batch size of 2048 and sequence length 512, resulting in a total of approximately 1T tokens seen by the model during pre-training (both unsupervised and supervised inclusive). We use the T5.1.1 architecture (Shazeer, 2020) for all of our experiments — which uses GEGLU-activated layers instead of ReLU in classic Transformer models (Vaswani et al., 2017). For optimization, we use Adafactor with an inverse square root learning rate schedule that kicks in after a a constant phase of 0.01 for 10k steps. ExT5 also uses the same tokenizer as T5.

**Fine-tuning**   We follow the same fine-tuning procedure for T5 and ExT5 for fair comparison, although we found that ExT5 generally benefitted from a smaller learning rate while fine-tuning ($10^{-4}$ worked well for ExT5 vs $10^{-3}$ for T5 variants). Fine-grained details can be found in Appendix B.

### 3.2 Experimental Setup

Our experiments consider both within-mixture and out-of-mixture tasks (i.e., whether a task is included in ExMix). Within-mixture tasks measure the amount the task benefits from multi-task pre-training and extreme task scaling. Similar to the co-trained models in Raffel et al. (2020), we continue to fine-tune on the target task from a pre-trained ExT5 checkpoint. For out-of-mixture tasks, we consider possibly new unseen tasks or collections that were not included in the ExMix mixture to test the effect of generalizing to unseen tasks. For the sake of brevity, the fine-grained details of these experimental setups can be found in the Appendix.

### 3.3 Experimental Results

#### Within-mixture Results

We report results on **SuperGLUE** (Table 5), **GEM** (Table 6), **Rainbow** (Table 7), **MsMarco** (Table 8) and **CBQA** datasets (Table 9). On the whole, we observe that ExT5 consistently outperforms strong T5 baselines across a range of model sizes. On SuperGLUE, we achieve $+5\%$, 2.3% and

+0.7% gain on BASE, LARGE and XL respectively. On GEM, ExT5 outperforms T5 on 6 out of 9 collections while remaining on-par on the other 3 collections. Notably, the gain on datasets such as WebNLG are approximately +11% ROUGE for the large model and generally range from +1% to +6% on different collections. On Rainbow, ExT5 outperforms our own run of T5 by +0.7% on average and +4.6% improvement over the best multi-task (sequential) setup in Lourie et al. (2021). Finally, on question answering and ranking, ExT5 substantially outperforms T5 at two different sizes.

| Model | BoolQ | CB | Copa | MultiRC | ReC | RTE | WiC | WSC | AVG |
|---|---|---|---|---|---|---|---|---|---|
| T5.1.1$_{BASE}$ | 82.3 | 91.7/92.9 | 60.0 | 76.9/39.6 | **80.9/80.2** | 84.5 | 69.3 | 81.7 | 76.1 |
| ExT5$_{BASE}$ | **82.6** | **98.7/98.2** | **73.0** | **79.5/45.4** | 80.8/80.0 | **87.0** | **71.3** | **83.7** | **79.9** |
| T5.1.1$_{LARGE}$ | 88.3 | 94.3/96.4 | 87.0 | 85.4/55.1 | **89.2/88.5** | 90.6 | **73.5** | 88.5 | 85.3 |
| ExT5$_{LARGE}$ | **88.4** | **98.7/98.2** | **89.0** | **85.5/58.0** | 88.6/87.9 | **93.1** | 73.4 | **96.2** | **87.3** |
| T5.1.1$_{XL}$ | **89.6** | 92.0/96.4 | 96.0 | **88.2/64.1** | **92.4/91.7** | 91.7 | 74.3 | **95.2** | 88.7 |
| ExT5$_{XL}$ | 89.4 | **100/100** | **97.0** | 87.5/62.7 | 91.4/90.9 | **94.2** | **74.6** | 93.3 | **89.4** |
| T5.1.1$_{XXL}$ | 90.4 | **100.0/100.0** | **99.0** | 88.6/63.9 | 91.0/90.1 | 92.1 | **78.5** | 95.2 | 90.2 |
| ExT5$_{XXL}$ | **91.1** | 94.9/96.4 | 98.0 | **89.4/66.0** | **93.3/92.7** | **95.7** | 77.3 | **96.2** | **90.6** |

Table 5: Comparisons of T5 and ExT5 on SuperGLUE validation sets.

| Model | Metric | WebNLG | DART | SGD | E2E | CG | ToTTo | WiA-A | WiA-T | WLE |
|---|---|---|---|---|---|---|---|---|---|---|
| T5.1.1$_{BASE}$ | METEOR | 0.323 | 0.364 | 0.325 | **0.383** | 0.201 | 0.366 | 0.302 | **0.368** | 0.189 |
|  | ROUGE-2 | 39.46 | 45.62 | 36.25 | **47.40** | 17.32 | 49.8 | 38.58 | **51.54** | 19.19 |
|  | BLEU | 29.06 | 34.75 | 33.44 | **43.17** | 8.34 | 39.59 | 29.53 | 42.71 | 14.72 |
| ExT5$_{BASE}$ | METEOR | **0.349** | **0.367** | **0.330** | 0.382 | **0.206** | **0.368** | **0.306** | 0.367 | **0.192** |
|  | ROUGE-2 | **45.07** | **46.87** | **37.46** | 47.32 | **18.13** | **50.17** | **39.10** | 51.35 | **19.41** |
|  | BLEU | **32.36** | **35.15** | **34.34** | 42.71 | **9.39** | **40.01** | **30.04** | **43.39** | **14.96** |
| T5.1.1$_{LARGE}$ | METEOR | 0.344 | 0.363 | 0.324 | **0.382** | 0.202 | 0.368 | **0.301** | **0.362** | 0.196 |
|  | ROUGE-2 | 43.31 | 45.22 | 36.17 | 46.60 | 17.01 | 49.90 | **38.37** | **50.52** | 20.47 |
|  | BLEU | 31.67 | 34.31 | 33.15 | **42.57** | 8.38 | 39.79 | **29.30** | **42.12** | 15.55 |
| ExT5$_{LARGE}$ | METEOR | **0.365** | **0.376** | **0.330** | 0.381 | **0.214** | **0.369** | 0.300 | 0.358 | **0.204** |
|  | ROUGE-2 | **48.17** | **48.14** | **37.77** | 46.70 | **19.04** | **50.33** | 37.98 | 50.38 | **21.16** |
|  | BLEU | **35.03** | **36.62** | **34.74** | 42.25 | **9.68** | **40.14** | 29.23 | 41.39 | **16.64** |

Table 6: Comparisons of T5 and ExT5 on GEM (English).

| Model | $\alpha$NLI | CosmosQA | HellaSwag | PIQA | SocialIQA | Winogrande | AVG |
|---|---|---|---|---|---|---|---|
| T5$_{LARGE}$ (multitask)[†] | 78.40 | 81.10 | 81.30 | 80.70 | 74.80 | 72.10 | 78.07 |
| T5$_{LARGE}$ (sequential)[†] | 79.50 | 83.20 | 83.00 | 82.20 | 75.50 | 78.70 | 80.35 |
| T5.1.1$_{LARGE}$ | **82.51** | 85.59 | 88.57 | **85.53** | 78.51 | 79.79 | 83.42 |
| ExT5$_{LARGE}$ | 82.25 | **85.86** | **88.99** | 85.04 | **79.73** | **82.53** | **84.07** |
| % Gain | -0.3% | +0.3% | +0.5% | -0.6% | +1.6% | +3.4% | +0.8% |

Table 7: Results on the Rainbow Commonsense Reasoning benchmark validation sets. Results with † are from Lourie et al. (2021).

| Model | MRR@10 |
|---|---|
| T5$_{LARGE}$ (Nogueira et al., 2020) | 0.393 |
| ExT5$_{LARGE}$ | **0.402** |
| % Gain | +2.3% |
| T5$_{XL}$ (Nogueira et al., 2020) | 0.398 |
| ExT5$_{XL}$ | **0.403** |
| % Gain | +1.3% |

Table 8: Results on MSMarco.

| Model | NQ | WQ | TQA |
|---|---|---|---|
| T5.1.1$_{LARGE}$ | 27.3 | 29.5 | 28.5 |
| ExT5$_{LARGE}$ | **28.6** | **30.5** | **30.7** |
| % Gain | +4.8% | +3.4% | +7.7% |
| T5.1.1$_{XL}$ | 29.5 | 32.4 | 36.0 |
| ExT5$_{XL}$ | **30.6** | **35.2** | **37.0** |
| % Gain | +3.7% | +8.6% | +2.8% |

Table 9: Results on CBQA dev sets.

OUT-OF-MIXTURE RESULTS

We are also interested in evaluating EXT5 on tasks outside of EXMIX, and hypothesize that the extreme multi-task pre-training of EXT5 will lead to better performance on new unseen settings. Concretely, we fine-tune and evaluate on **Machine Translation**: translating sentences from English to other languages (Bojar et al., 2014; 2015; 2016); **Reasoning**: answering scientific questions on ARC (Clark et al., 2018); and **Named Entity Recognition**: extracting all entities from sentences on the CoNLL-2003 NER dataset (Tjong Kim Sang & De Meulder, 2003).

| Model | Machine Translation | | | QA | NER | |
|---|---|---|---|---|---|---|
| | EnDe | EnFr | EnRo | ARC | Dev | Test |
| Raffel et al. (2020) | 26.98 | 39.82 | 27.65 | - | - | - |
| T5.1.1$_{\text{BASE}}$ | 28.30 | 41.01 | 28.11 | 26.45 | 92.55 | 85.75 |
| ExT5$_{\text{BASE}}$ | **28.32** | **41.89** | **28.38** | **36.35** | **92.68** | **86.53** |
| % Gain | ±0% | +2.1% | +1.0% | +37.4% | +0.13% | +0.91% |
| T5.1.1$_{\text{LARGE}}$ | 28.68 | 41.34 | 29.01 | 55.80 | 92.80 | 86.56 |
| ExT5$_{\text{LARGE}}$ | **28.98** | **42.71** | **29.49** | **63.99** | **93.63** | **87.34** |
| % Gain | +1.0% | +3.3% | +1.7% | +14.7% | +0.90% | +0.90% |

Table 10: Experimental results on tasks that are not in EXMIX. For ARC, we report test scores on the challenge set with retrieval. For NER, we report accuracy on a sentence level (see Appendix B.2).

Table 10 summarizes the results on the out-of-mixture tasks. Across all tasks, we see that EXT5 outperforms upon T5 baselines. The largest improvement is on the ARC scientific reasoning task, perhaps due to the large amount of QA tasks in EXMIX. Though, the trend is consistent also with the NER and MT tasks that do not have any similar dataset in EXMIX. This suggests that the representations learned by EXT5 are more general adaptable to a new objective, even when the output is in a new language.

This improved generalization of EXT5 is very encouraging from a practical stand point, since pre-training again with EXMIX ∪ {$t$} for any new target task $t$ would be very expensive. Instead, we see that the extreme multi-task pre-training of EXT5 already provides improved results. Therefore, it might only be worth repeating pre-training when the collection of training datasets grows by a significant amount (see §2.5).

## 4 RELATED WORK

**Improving NLP models with Multi-task Learning**   Collobert & Weston (2008) leverage multi-task learning for relatively simple tasks like Part-of-Speech tagging. Phang et al. (2019) use an intermediate fine-tuning stage using four tasks with large datasets for Natural Language Understanding. Similarly, Liu et al. (2019a) proposed MT-DNN, which uses a setup at a scale of around 30 tasks and up to 440M parameters. Most recently, Aghajanyan et al. (2021) use around 50 tasks and models of sizes upto 440M parameters. Gururangan et al. (2020) take an alternative approach, which is to continue pre-training a model but use domain-specific data as an intermediate step. McCann et al. (2018) proposed a unified framework similar to that of T5. Recently, Wei et al. (2021) also illustrated how a multi-task learning stage can greatly improve the zero-shot prompting performance of large language models at the scale of ~137B parameters. Efforts have also been made to tailor pre-training objectives to specific tasks, e.g., question answering (Ram et al., 2021; Jia et al., 2021), dialogue (Li et al., 2020), and span selection tasks (Joshi et al., 2020).

**Relationships amongst different tasks**   Bingel & Søgaard (2017) conducted a study similar to ours in §2.1 but for more traditional NLP tasks like chunking, CCG tagging, POS tagging, etc. More recently, Vu et al. (2020) conducted an in-depth study of relationships between various classification/regression, question-answering, and sequence-labeling tasks, and proposed a task-embedding framework to predict such relationships. Khashabi et al. (2020) also conducted similar experiments but specific to question-answering datasets/formats, resulting in a strong QA model known as UnifiedQA that is also based on the T5 framework. Outside of NLP, Zhang & Yeung (2010) introduced a convex optimization objective for *learning* task relationships, and Li et al. (2018) explore and exploit task relationships on a variety of diverse datasets.

**Choosing which tasks to transfer from**    Our experiments in §2.2 attempted to empirically select a set of tasks to transfer from. Along these lines, Ruder & Plank (2017) use a Bayesian Optimization method with similarity measures to automatically select relevant data from different domains. Similarly, Guo et al. (2019) use multi-armed bandits to select tasks and a Gaussian Process to control the mixing rates for the selected tasks. Another strand of recent work selects appropriate transfer languages based on manually defined features (Lin et al., 2019; Sun et al., 2021). Aside from the NLP domain, Fifty et al. (2021) proposed a method to select which tasks to transfer to based on task gradients. All of the aforementioned works select data tailored to a downstream task of interest. If a general pre-trained model was attempted to be trained in a similar fashion, computational bottlenecks similar to those motivating §2.1 and §2.2 would arise.

**Pre-trained Transformers**    Transformer models (Vaswani et al., 2017) such as T5 (Raffel et al., 2020), BERT (Devlin et al., 2019) and GPT-3 (Brown et al., 2020) rely on large unlabeled corpus for self-supervised learning. Given the wild success of the pre-train-finetune paradigm, the search for suitable pre-training tasks has also become an active area of research (Lewis et al., 2019; Lan et al., 2019; Chang et al., 2020; Zhang et al., 2019; Lourie et al., 2021). While there has been evidence that supplementary pre-training tasks can help improve performance, this work is the first massive-scale multi-task pre-trained model.

**Scaling Laws**    Scaling laws for Transformers have attracted much attention recently, especially pertaining to model size (Kaplan et al., 2020; Zhai et al., 2021; Tay et al., 2021a). In Kaplan et al. (2020), the authors further investigate scaling with respect to dataset size (on the same pre-training corpus). To this end, this work can be interpreted as an attempt of scaling up with respect to the number of high quality, diverse labeled tasks that can be used for pre-training.

## 5    EPILOGUE

**Limitations**    Despite our best efforts to evaluate on as many representative tasks as possible while also maintaining a balance among task partitions for a given set of transfer learning experiments, any study that explicitly abstracts datasets into "task families" is highly dependent on nuances pertaining to the nature, domain, and expressiveness of the task family's representative datasets. For this paper, the subsets were constructed so as to include a diverse set of datasets to evaluate on, and we tried to partition task-families to be as mutually exclusive as possible. However, it must be acknowledged that no dataset is perfectly isolated, and any set of them only a proxy for a larger "task family". On a separate note, lexical metrics like BLEU/ROUGE are useful but do not paint the full picture of how well a model truly performs on text-generation tasks.

**Future Work**    We believe that a multilingual version of EXT5 would be a natural extension of this work. Such a model will require extra care with regard to balancing not only task families, but also task languages. A multilingual version of EXMIX could provide a more robust foundation for the analysis of task families in existing works that analyze how multilingual NLP models transfer amongst different languages (Kudugunta et al., 2019; Hu et al., 2020; Wang et al., 2021). For example, it would be interesting to understand whether our results in §2.1 hold across different languages (and language families), and to explore cross-lingual cross-task generalization. We also hypothesize that modeling innovations that introduce inductive biases designed to exploit multi-task learning setups (Ha et al., 2016; Tay et al., 2021b) can push the boundary of the strong performance displayed by EXT5. Other solutions like gradient manipulation (Yu et al., 2020; Wang et al., 2021) might also further improve extreme multi-task scaling, albeit at the cost of more complex implementations.

**Conclusion**    This paper explores how supervised multi-task learning at a massive scale can be used to improve existing self-supervised pre-training strategies for NLP models, and does so by introducing EXMIX (§2) and EXT5 (§3). Our experiments showed that while negative transfer is common when fine-tuning on diverse tasks (§2.1), scaling up the number of tasks to include in a multi-task pre-training setup enables strong downstream performance (§3.2) with better sample-efficiency (§2.6). We hope that this paper motivates future research on how existing labeled datasets can be used to further improve NLP models within the pre-train/fine-tune paradigm.

## ACKNOWLEDGEMENTS

The authors would like to thank Mostafa Dehghani and Adhi Kuncoro for valuable comments and insights. We would also like to thank the authors of Mesh Tensorflow (Shazeer et al., 2018) and T5 (Raffel et al., 2020), as their high-quality code and paper enabled this work.

## AUTHOR CONTRIBUTIONS

**Vamsi** co-led the project and was primarily responsible for the design, implementation, and engineering effort behind it. Vamsi proposed and ran key experiments including (but not limited to): task-family transfer analysis, early proof-of-concepts for extreme task-scaling and EXT5, task-scaling analysis, etc. Vamsi also wrote most of the paper, and was (jointly) responsible for the overall framing of the paper.

**Yi** served as the co-lead of the paper and proposed the initial idea. Yi was responsible for most of the downstream EXT5 experiments, including SuperGLUE, Rainbow, CBQA, and Machine translation, along with large-scale pre-training of EXT5. Yi also wrote large portions of the paper.

**Tal** was (jointly) responsible for the overall framing of the paper, and wrote large portions of it. Tal contributed the Vitamin C task to the EXMIX mixture, along with running out-of-mixture experiments for Named Entity Recognition.

**Jinfeng** contributed the GEM benchmark to our mixture and was heavily involved in running experiments.

**Steven** contributed the DialoGLUE tasks to EXMIX, helped run a substantial number of experiments and contributed substantially to discussions around the paper's framing.

**Sanket** was responsible for and ran experiments on GEM, and contributed the Yahoo Answers and Argument mining tasks.

**Honglei** contributed the MsMarco task to EXMIX and helped with benchmarking EXT5 on Ms-Marco. Honglei also contributed scripts and pipelines for obtaining reranking results on MsMarco.

**Vinh** contributed NewsQuiz, AgreeSum, TweetQa and TweetEval to EXMIX and contributed substantially to paper writing and the framing of the paper.

**Dara** contributed GPT DeepFake tasks and low-resource tasks (which were not used in the end).

**Jianmo** contributed the DocNLI task and helped with running experiments for out-of-mixture tasks.

**Jai** contributed the Race and MultiRC Eraser tasks to the EXMIX and helped edit the paper.

**Kai** contributed several retrieval tasks to EXMIX, which were not included eventually.

**Sebastian** helped substantially with the paper narrative, writing of the paper and brainstorming.

**Donald** (along with Yi and Vamsi) was heavily involved in framing the early vision of the project. Donald was also deeply involved in brainstorming sessions, and provided critical feedback that helped to steer the project in the right direction.

*All authors contributed to brainstorming and discussion sessions.*

## ETHICS STATEMENT

Large language models have been shown to capture certain biases about the data they have been pre-trained on (Bender et al., 2020). While a comprehensive analysis of such biases is outside of the scope of this work, it is a compelling direction to investigate to what extent the inclusion of supervised data during pre-training can help mitigate such biases. An alternative consideration is the addition of diverse values-targeted data (Solaiman & Dennison, 2021) during pre-training in order to instill beneficial biases in a model.

Another challenge when training large models is their energy consumption and environmental impact (Strubell et al., 2019). To ablate different task combinations, we performed experiments using

the more computationally efficient fine-tuning setup. We have shown that EXMIX leads to more sample-efficient pre-training compared to standard self-supervision, which we hope will save compute in future experiments.

## REPRODUCABILITY STATEMENT

All of the modeling and training code used for ExT5 and its variants is already open-sourced as a part of the Mesh Tensorflow[1] (Shazeer et al., 2018) and T5[2] (Raffel et al., 2020) Libraries. Additionally, EXMIX is composed of datasets that are already publicly available.

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

# A  DATASETS

| Dataset(s) | Description | No. Train Datasets | $|\mathcal{D}|$ | Citation |
|---|---|---|---|---|
| GLUE | General Language Understanding | 7 | 949,101 | Wang et al. (2019b) |
| SuperGLUE | General Language Understanding | 8 | 185,673 | Wang et al. (2019a) |
| KILT | Knowledge-Intensive Language Tasks | 9 | 3,129,859 | Petroni et al. (2021) |
| Rainbow | Commonsense Reasoning | 6 | 324,742 | Lourie et al. (2021) |
| GEM (en) | Natural Language Generation | 8 | 1,067,955 | Gehrmann et al. (2021) |
| DialoGLUE | Dialogue Understanding | 6 | 76,122 | Mehri et al. (2020) |
| TweetEval | Twitter Classification Benchmark | 8 | 120,104 | Barbieri et al. (2020) |
| CNN/Dailymail | News Summarization | 1 | 287,113 | See et al. (2017) |
| XSum | News Summarization | 1 | 203,577 | Narayan et al. (2018) |
| Multi-News | News Summarization | 1 | 44,972 | Fabbri et al. (2019) |
| AESLC | Email Summarization | 1 | 14,436 | Zhang & Tetreault (2019) |
| Gigaword | Summarization | 1 | 3,803,957 | Rush et al. (2015) |
| SamSum | Dialogue Summarization | 1 | 14,372 | Gliwa et al. (2019) |
| ANLI | Adverserial NLI | 1 | 162,865 | Nie et al. (2020) |
| ESNLI | Explainable NLI | 1 | 549,367 | DeYoung et al. (2020) |
| AgreeSum Entailment | Article-Summary NLI | 1 | 7,750 | Pang et al. (2021) |
| DocNLI | Document NLI | 1 | 942,314 | Yin et al. (2021) |
| Vitamin C | Fact-checking NLI | 1 | 370,653 | Schuster et al. (2021) |
| Web Questions | QA (open) | 1 | 3778 | Berant et al. (2013) |
| SQuAD | QA (context) | 1 | 87,599 | Rajpurkar et al. (2016) |
| QuAC | QA (context) | 1 | 83,568 | Choi et al. (2018) |
| DROP | QA (Discrete Reasoning) | 1 | 77,409 | Dua et al. (2019) |
| RACE | School QA (MCQ) | 4 | 113,013 | Lai et al. (2017) |
| Eraser MultiRC | Explainable QA (MCQ) | 1 | 24,029 | DeYoung et al. (2020) |
| TweetQA | QA (context) | 1 | 10692 | Xiong et al. (2019) |
| NewsQuizQA | Question-Answer Generation | 1 | 16,160 | Lelkes et al. (2021) |
| Amazon Reviews | Review Classification | 1 | 100,000 | ama |
| GoEmotions | Emotion Classification | 1 | 43.410 | Demszky et al. (2020) |
| IMDb Reviews | Sentiment Classification | 1 | 25,000 | Maas et al. (2011) |
| Sentiment140 | Sentiment Classification | 1 | 1,600,000 | Go et al. (2009) |
| Yelp Reviews | Sentiment Classification | 1 | 560,000 | Zhang et al. (2015) |
| AGNews | News Classification | 1 | 120,000 | Zhang et al. (2015) |
| TreqQC | Question Classification | 1 | 5000 | Hovy et al. (2001) |
| Civil Comments | Toxicity Classification | 1 | 1,804,874 | Borkan et al. (2019) |
| Wiki Toxicity | Toxicity Classification | 1 | 159,571 | Wulczyn et al. (2017) |
| Yahoo! Answers | Topic Classification | 1 | 140,000 | Zhang et al. (2015) |
| UKP Arg. Mining | Argument Classification | 1 | 18,341 | Stab et al. (2018) |
| Parsing to FunQL | Semantic Parsing | 3 | 5,565 | Guo et al. (2020) |
| Parsing to interm. repr. | Semantic Parsing | 4 | 117,318 | Herzig et al. (2021) |
| COGS | Semantic Parsing (Comp. Gen.) | 1 | 24,155 | Kim & Linzen (2020) |
| GPT Deepfake detection | Generated-text classification | 8 | 500,000 | Radford et al. (2019) |
| StylePTB | Style Transfer | 4 | 53,546 | Lyu et al. (2021) |
| Shakespearizing English | Style Transfer | 2 | 36,790 | Jhamtani et al. (2017) |
| MS-MARCO | Pointwise Ranking | 1 | 100,000 | Bajaj et al. (2018) |
| Total | EXMIX | 107 | 18,085,040 | - |

Table 11: All of the training datasets used to construct ExMix.

Table 11 summarizes the 107 datasets included in EXMIX. Some of the lines in the table represent existing benchmarks that group several tasks together. From each collection, we use the datasets that include English training data:

- **GLUE**: CoLA (Warstadt et al., 2019), SST-2 (Socher et al., 2013), MRPC (Dolan & Brockett, 2005), QQP, STS-B (Cer et al., 2017), MNLI (Williams et al., 2018), QNLI (Converted from Rajpurkar et al. (2016), RTE (Dagan et al., 2006), WNLI (Sakaguchi et al., 2020).

- **SuperGLUE**: BoolQ (Clark et al., 2019), CB (De Marneffe et al., 2019), COPA (Roemmele et al., 2011), MultiRC (Khashabi et al., 2018), ReCoRD (Zhang et al., 2018), RTE (Dagan et al., 2006), WiC (Pilehvar & Camacho-Collados, 2019), WSC (Levesque et al., 2011).

- **KILT**: FEVER (Thorne et al., 2018), AIDA (Yosef et al., 2011), WNED (Guo & Barbosa, 2014), T-REx (Guo & Barbosa, 2014), NQ (Kwiatkowski et al., 2019), HoPo (Yang et al., 2018), TQA (Joshi et al., 2017), ELI5 (Fan et al., 2019), WoW (Dinan et al., 2019).

- **Rainbow**: $\alpha$NLI (Bhagavatula et al., 2020), CosmosQA (Huang et al., 2019), HellaSWAG (Zellers et al., 2019), PIQA (Bisk et al., 2020), SocialIQA (Sap et al., 2019), WinoGrande (Sakaguchi et al., 2021).

- **GEM (en)**: Wiki-Lingua (Faisal Ladhak & McKeown, 2020), WenNLG (Gardent et al., 2017; Castro Ferreira et al., 2020), CommonGEN (Lin et al., 2020), E2E (Dušek et al., 2019), DART

(Radev et al., 2020), ToTTo (Parikh et al., 2020), Wiki-Auto (Jiang et al., 2020), TurkCorpus(Xu et al., 2016)

– **DialoGLUE**: Banking77 (Casanueva et al., 2020), HWU64 (Liu et al., 2019b), CLINC150 (Larson et al., 2019), SGD (Rastogi et al., 2020), TOP (Gupta et al., 2018).

– **TweetEval**: Emotion Recognition (Mohammad et al., 2018), Emoji Prediction (Barbieri et al., 2018), Irony Detection (Van Hee et al., 2018), Hate Speech Detection (Basile et al., 2019), Offensive Language Identification (Zampieri et al., 2019), Sentiment Analysis (Rosenthal et al., 2017), Stance Detection (Mohammad et al., 2016).

## B    EXPERIMENTAL DETAILS

This section describes the experimental details

### B.1    IMPLEMENTATION DETAILS

Our models were trained using Mesh Tensorflow (Shazeer et al., 2018) using the T5 library (Raffel et al., 2020).

### B.2    DATASET EXPERIMENTAL SETUP

This section reports the dataset and experimental setup on each individual target task/dataset.

**SuperGLUE**    We finetune on the entire SuperGLUE as a mixture with proportionate sampling in similar fashion to (Raffel et al., 2020). We finetune for a total of 200k steps with a batch size of 128. When selecting checkpoints on SuperGLUE, we follow the same convention as Raffel et al. (2020) in selecting the best checkpoint for each task for a fair comparison to models that are fine-tuned on the individual tasks instead of co-training on all of them.

**GEM**    We report test set results on all datasets except CommonGen and ToTTo, on which we report validation scores. We sweep over learning rates of $10^{-3}$, $5 \times 10^{-4}$ and $10^{-4}$. All results are computed using GEM-metrics[3]. For each dataset, we select the best model checkpoint using average of BLEU, ROUGE-1, ROUGE-2 and ROUGE-L scores on the validation set. We use the greedy decoding strategy to be consistent with the original GEM paper (Gehrmann et al., 2021).

**CBQA**    We report validation set results, and sweep over learning rates of $10^{-3}$ and $10^{-4}$.

**Rainbow**    We multi-task co-train on all datasets, and sweep over learning rates of $10^{-3}$ and $10^{-4}$.

**WMT Machine Translation**    We finetune our models on three collections of WMT, namely EnDe, EnFr and EnRo. We use a constant learning rate of $10^{-3}$ and dropout of 0.1. We train with a batch size of 4096 for a maximum of $400k$ steps and report peak validation BLEU score. We use a beam size of 4 and a length penalty of 0.6.

**ARC**    We report scores on the Challenge set, and train with a batch size of 32 and sweep over learning rates of $10^{-3}$ and $10^{-4}$.

**CoNLL-03 NER**    We convert NER to seq2seq by writing the target as the ordered sequence of tags and entities (for example *"When Alice visited New York"* → *"[PER] Alice [LOC] New York"*). Accuracy is measured on a sentence level, considering a prediction to be correct only if it exactly matches the reference sequence.

## C    DETAILED EXPERIMENTAL RESULTS

Many of our experiments in §2 used the average SuperGLUE score of a model for evaluation. We report the full results on all datasets below.

---

[3]https://github.com/GEM-benchmark/GEM-metrics

| Mixture | BoolQ | CB | Copa | MultiRC | ReC | RTE | WiC | WSC | AVG |
|---|---|---|---|---|---|---|---|---|---|
| Vanilla | 82.3 | 91.7/92.9 | 60.0 | 76.9/39.6 | 80.9/80.2 | 84.5 | 69.3 | 81.7 | 76.1 |
| Best-effort | 81.7 | 89.4/92.9 | 75.0 | 76.6/37.4 | 76.4/75.5 | 82.7 | 67.1 | 80.8 | 76.4 |
| Random-55 | 81.3 | 97.3/97.0 | 67.7 | 77.0/39.7 | 76.5/75.6 | 82.7 | 69.5 | 83.3 | 77.0 |
| ExT5 | 82.6 | 98.7/98.2 | 73.0 | 79.5/45.4 | 80.8/80.0 | 87.0 | 71.3 | 83.7 | **79.9** |

Table 12: Full SuperGLUE results from §2.2

| Method | BoolQ | CB | Copa | MultiRC | ReC | RTE | WiC | WSC | AVG |
|---|---|---|---|---|---|---|---|---|---|
| Vanilla | 82.3 | 91.7/92.9 | 60.0 | 76.9/39.6 | 80.9/80.2 | 84.5 | 69.3 | 81.7 | 76.1 |
| Pre-finetuning | 82.2 | 85.1/89.3 | 74.0 | 79.8/45.1 | 79.2/78.3 | 87.7 | 69.6 | 82.7 | 78.1 |
| Multi-task pre-training | 82.6 | 98.7/98.2 | 73.0 | 79.5/45.4 | 80.8/80.0 | 87.0 | 71.3 | 83.7 | **79.9** |

Table 13: Full SuperGLUE results from §2.3

| $R$ | BoolQ | CB | Copa | MultiRC | ReC | RTE | WiC | WSC | AVG |
|---|---|---|---|---|---|---|---|---|---|
| 0 | 76.8 | 58.6/83.9 | 63.0 | 66.6/22.6 | 53.0/52.1 | 75.5 | 63.2 | 73.1 | 65.0 |
| 1 | 82.1 | 93.7/94.6 | 75.0 | 78.0/41.7 | 76.7/75.7 | 85.6 | 68.8 | 76.9 | 77.3 |
| 2 | 82.6 | 98.7/98.2 | 73.0 | 79.5/45.4 | 80.8/80.0 | 87.0 | 71.3 | 83.7 | **79.9** |
| 4 | 81.3 | 96.0/94.6 | 73.0 | 75.2/38.8 | 77.4/76.6 | 84.8 | 68.8 | 83.7 | 77.6 |
| 5 | 81.9 | 89.4/92.9 | 74.0 | 75.5/35.6 | 76.2/75.3 | 85.6 | 69.1 | 76.9 | 76.2 |
| 10 | 81.2 | 93.2/96.4 | 77.0 | 75.6/37.6 | 76.5/75.6 | 82.7 | 70.4 | 80.8 | 77.4 |
| 20 | 80.7 | 93.7/94.6 | 71.0 | 74.5/36.5 | 75.9/74.4 | 79.8 | 66.5 | 84.6 | 75.9 |
| $\to \infty$ | 82.3 | 91.7/92.9 | 60.0 | 76.9/39.6 | 80.9/80.2 | 84.5 | 69.3 | 81.7 | 76.1 |

Table 14: Full SuperGLUE results from §2.4

| # Tasks | BoolQ | CB | Copa | MultiRC | ReC | RTE | WiC | WSC | AVG |
|---|---|---|---|---|---|---|---|---|---|
| | | | | **Batch Size = 128** | | | | | |
| 0 | 79.3 | 92.4/92.9 | 72.0 | 74.2/32.8 | 74.9/73.9 | 79.8 | 70.2 | 81.7 | 75.4 |
| 30 (random) | 78.7 | 95.7/95.2 | 66.0 | 72.6/30.5 | 72.8/72.0 | 77.6 | 68.4 | 82.4 | 74.1 |
| 55 (random) | 79.4 | 93.2/94.6 | 74.3 | 73.6/33.8 | 74.1/73.2 | 80.7 | 68.8 | 82.1 | 75.8 |
| 80 (random) | 80.0 | 92.5/94.6 | 70.0 | 74.3/34.8 | 73.9/72.9 | 80.0 | 68.1 | 82.4 | 75.3 |
| 107 | 79.3 | 95.0/96.4 | 74.0 | 74.0/34.8 | 73.5/72.5 | 79.4 | 70.7 | 77.9 | 75.6 |
| | | | | **Batch Size = 512** | | | | | |
| 0 | 82.3 | 91.7/92.9 | 60.0 | 76.9/39.6 | 80.9/80.2 | 84.5 | 69.3 | 81.7 | 76.1 |
| 30 (random) | 80.6 | 93.6/95.8 | 67.7 | 74.6/35.4 | 75.7/74.7 | 81.2 | 68.6 | 83.3 | 75.8 |
| 55 (random) | 81.3 | 97.3/97.0 | 67.7 | 77.0/39.7 | 76.5/75.6 | 82.7 | 69.5 | 83.3 | 77.0 |
| 80 (random) | 82.1 | 94.4/95.8 | 71.7 | 76.8/39.4 | 77.0/76.1 | 84.7 | 69.2 | 83.0 | 77.6 |
| 107 | 82.6 | 98.7/98.2 | 73.0 | 79.5/45.4 | 80.8/80.0 | 87.0 | 71.3 | 83.7 | **79.9** |

Table 15: Full SuperGLUE results from §2.5

| # Pre-train steps | BoolQ | CB | Copa | MultiRC | ReC | RTE | WiC | WSC | AVG |
|---|---|---|---|---|---|---|---|---|---|
| | | | | **T5.1.1** | | | | | |
| 20k | 77.9 | 93.0/92.9 | 69.0 | 73.0/32.3 | 73.3/72.4 | 77.6 | 69.6 | 79.8 | 73.8 |
| 50k | 82.3 | 100.0/100.0 | 74.0 | 76.8/38.0 | 79.9/79.1 | 82.3 | 70.4 | 83.7 | 78.7 |
| 100k | 83.7 | 95.0/96.4 | 82.0 | 80.0/45.9 | 83.8/83.0 | 87.0 | 73.7 | 84.6 | 81.6 |
| 200k | 85.7 | 100.0/100.0 | 85.0 | 81.8/49.0 | 85.2/84.4 | 87.7 | 73.5 | 88.5 | 83.8 |
| | | | | **ExT5** | | | | | |
| 20k | 80.3 | 95.0/96.4 | 70.0 | 74.4/35.8 | 72.9/72.1 | 82.7 | 68.5 | 81.7 | 75.8 |
| 50k | 83.1 | 97.4/96.4 | 78.0 | 79.2/43.9 | 79.6/78.9 | 88.1 | 73.4 | 87.5 | 81.0 |
| 100k | 85.3 | 100.0/100.0 | 81.0 | 81.6/48.9 | 83.7/83.0 | 89.2 | 73.2 | 90.4 | 83.5 |
| 200k | 86.5 | 98.7/98.2 | 86.0 | 83.2/53.1 | 85.4/84.7 | 91.7 | 73.4 | 93.3 | 85.3 |

Table 16: Full SuperGLUE results from §2.6

