# OpenReview forum: "ExT5: Towards Extreme Multi-Task Scaling for Transfer Learning"
_ICLR.cc/2022/Conference — ICLR 2022 Poster_

### Official Review · Reviewer_6JD9 · 2021-11-02

**Correctness:** 4
**Technical Novelty And Significance:** 2
**Empirical Novelty And Significance:** 3
**Recommendation:** 8
**Confidence:** 5

**Main Review:**

Pros
* The scale of the experiments is very impressive and the results are very strong - ExT5 is generally a much stronger model than it's T5 counterpart.
* From a resource perspective, both the pre-trained ExT5 model and the ExMix collection would be valuable to the NLP research community.
* Because of how well the pre-training of ExT5 and T5 are controlled to be as similar as possible, the argument that there is significant benefit to supervised pre-training is convincing.
* The analysis is helpful in demonstrating the role of the number of tasks in the final model performance. There is a clear relationship between the number of tasks used during pre-training and the strength of the downstream model, which motivates the scale of ExMix.
* The authors take a principled approach to selecting a subset of tasks and demonstrate that it is generally not successful over increasing the number of tasks, highlighting the importance of the number of tasks.
* Section 2.3 provides strong evidence that multi-task pre-training is stronger than multi-task pre-finetuning.
* Section 2.6 demonstrates that multi-task pre-training not only results in stronger final models than vanilla pre-training, but that throughout the entirety of training multi-task pre-training beats vanilla pre-training.

Cons
* The methodological novelty of the paper is fairly low. ExMix is a collection of pre-existing datasets and using multi-task signals during pre-training is not new either. The scientific value of this work predominantly comes from the scale of their multi-task setting, and their analyses. It is perhaps not surprising that a larger collection of supervised pre-training datasets would improve performance over these baselines.
* All of section 2 is evaluated on average SuperGLUE performance only. While SuperGLUE is a strong baseline given the diverse set of tasks it encompasses, there is always the possibility that a subset of tasks within SuperGLUE are particularly well-suited for the ExT5 training setup and, as such, Section 2 risks overfitting to the SuperGLUE baseline.
* There are some details missing regarding Section 2.4, especially with respect to random seeds. In particular, Figure 3 shows some interesting behavior regarding the hyperparameter R - it is surprising to me that performance does not monotonically decrease after a certain point but rather seems to be trending upwards. What happens if R increases even more? Or is the behavior at the tail end of R (R=5 or 10) due to random noise?
* Section 2.5 compares different numbers of tasks trained on different batch-sizes. However, the models use the same R parameter across numbers of tasks, which seems unfair to low-task settings given that it was tuned on 107 tasks. Additionally, the models use the same learning rate regardless of batch-size, ostensibly a learning rate that was tuned for a batch-size of 512. It has been previously noted (e.g. by McCandlish et al., 2018) that there is a clear relationship between batch-size and learning rate; thus, if the same learning rates are being used across the 512 and 128 batch-size settings, it's likely that the results are heavily influenced by a poor learning rate for the 128 batch-size setting.

**Summary Of The Paper:**

This paper introduces "ExMix," a collection of 107 pre-existing NLP datasets across 8 identified task-families. ExMix is proposed as a pre-training dataset, to be mixed with the typical self-supervised pre-training prevalent in NLP. The authors propose to pre-train a T5 model on a mixture of ExMix and C4 (the self-supervised pre-training dataset for T5), which they term ExT5. To do this, they convert every task to a text-to-text problem, so that all tasks, supervised and self-supervised, use standard sequence-to-sequence cross-entropy loss. They pre-train ExT5 for the same number of steps and on the same number of tokens as T5. They then evaluate how ExT5 and T5 perform when fine-tuned to individual tasks. They find that ExT5 largely outperforms T5 on the majority of downstream evaluated tasks, including tasks whose training data was included in ExMix as well as tasks which were not seen during pre-training.

The authors perform an experiment to identify whether or not a manually curated subset of ExMix can provide better pre-training generalization than the full set of tasks, due to negative transfer between tasks. To do this, they examine negative transfer between pairs of task families during fine-tuning. They select the task-families which exhibited the highest amount of fine-tuning transfer across all task-pairs, and use only 48 tasks from these task-families as a pre-training objective. They compare this model to a model trained on a slightly larger subset (55) of randomly selected tasks, as well as ExT5 (trained on all 107) tasks. The authors find that their method of task-selection is largely unhelpful for selecting pre-training tasks - the model trained on a random subset of 55 tasks outperforms the manually curated 48-task model, and ExT5 significantly outperforms both. This suggests that, for pre-training, selecting tasks that complement one-another with respect to downstream performance is not straight-forward, but increasing the number of tasks seems to alleviate negative transfer, rather than exacerbate it.

Finally, the authors perform some additional analysis of the training decisions made around ExT5. They find that: (a) pre-finetuning, which introduces multi-task learning as an intermediate step between self-supervised pre-training and fine-tuning, is not better than mixing all tasks for pre-training when using ExT5, (b) ExMix without self-supervised pre-training is significantly worse than self-supervised training without ExMix, suggesting that self-supervised pre-training is still crucial for pre-training, (c) downstream performance generally increases as the number of tasks scales up, and (d) pre-training with ExMix is significantly more sample-efficient than self-supervised pre-training along, i.e. it achieves strong downstream results very early, even 20k steps into pre-training.

**Summary Of The Review:**

Overall, the paper fairly convincingly argues that mixing supervised and self-supervised signals during pre-training is extremely beneficial to downstream performance and sample-efficiency. However, there is limited methodological novelty and the benefits of scale and incorporating supervision may not be particularly surprising.

---

> ### Author Response · Authors · 2021-11-16
> **Response to Reviewer 6JD9**
>
> We thank the reviewer for their feedback and suggestions. We attempt to address their concerns below:
>
> ### Is it surprising that extreme multi-task scaling works?
> We believe that it is not obvious that extreme multi-task scaling would work. In the original T5 paper ([Raffel et al.](https://arxiv.org/abs/1910.10683)), the authors found that including ~20 tasks in their multi-task pre-training setup did not significantly affect results (see section 3.5.3 in [Raffel et al.](https://arxiv.org/abs/1910.10683)). The conclusion of [Raffel et al.](https://arxiv.org/abs/1910.10683) (page 37) was “pre-training on a multi-task mixture of unsupervised and supervised tasks before fine-tuning worked as well as pre-training on the unsupervised task alone”. In fact, they later proposed T5.1.1, which does not use any supervised data while pre-training.
>
> This paper challenges that assertion, and shows that multi-task pre-training does indeed help when scaled. Moreover, [Aghajanyan et al.](https://arxiv.org/abs/2101.11038) had to modify the loss ([R3F/R4F](https://openreview.net/forum?id=OQ08SN70M1V)) for robust fine-tuning to get the model to perform well. [Aghajanyan et al.](https://arxiv.org/abs/2101.11038) and [Liu et al.](https://arxiv.org/abs/1901.11504) also require task-specific heads and use more training steps than pre-trained models. Our model requires none of these, but rather shares all parameters across all tasks thanks to the seq2seq design. This allows us to directly measure the impact of multi-task scaling and have a lower risk of conflating it with other tweaks to model setup and more training steps.
>
> ### Tail behavior of R
> We are running experiments using R=20 and plan on including them in our revised manuscript (before the discussion period ends)
>
> ### Average SuperGLUE performance
> We agree with the reviewer here, and will include all of the full SuperGLUE tables in the appendix in our revised submission (to be uploaded before the discussion period ends). However, we also note that our final downstream evaluations use a wide variety of tasks that ExT5 performs well on, so we find it unlikely that our conclusions overfit to SuperGLUE.
>
> ### Tailoring R to different task scales
> We agree with the reviewer here, but unfortunately our experiments consider 3 pre-training runs per random seed (of task selection) per batch size, resulting in 3x3x2=18 total runs, making an additional sweep for ideal values of R computationally expensive.
> However, we offer a potential explanation as to why R=2 is still a strong hyperparameter: R=1 also works well for our set of 107 tasks, which gives us reason to believe that decreasing the number of tasks at R=2 would also be a strong model. In other words, the fraction of supervised examples included with `{N_TASKS=107, R=1}` is similar to `{N_TASKS in (30,55,80), R=2}`, meaning R=2 likely still results in a strong model for different task sizes.
>
> ### Tailoring learning rate to different batch size
> While we agree that the learning rate could have been tuned for smaller batch sizes, we note that the original T5 paper ([Raffel et al.](https://arxiv.org/abs/1910.10683)) also used the exact same hyperparameters for their analysis (see sec 3.1.2 of their paper), which resulted in strong performance at a batch size of 128. We also note that the results using a batch size of 128 are still strong (using 55 random tasks with B=128 matches using 0 tasks at B=512, which is unusual given that pre-training almost always benefits from the largest batch size possible).
>
> *Please let us know if our answers and clarifications addressed your concerns. We welcome any other comments or suggestions that will help us further improve the paper.*

---

### Official Review · Reviewer_4oQF · 2021-11-03

**Correctness:** 4
**Technical Novelty And Significance:** 2
**Empirical Novelty And Significance:** 3
**Recommendation:** 6
**Confidence:** 4

**Main Review:**

strength:
- The paper is well-motivated and clearly presented;

- The authors extensively study the effect of co-training, multi-task learning at scale by leveraging multi-task learning naively on top of pre-trained representation, further testing on multi-task pretraining and pre-finetuning, which suggests a practical way to leverage multi-task learning and how should that improve the sample efficiency of the learned model;

- Substantial ablations are provided for understanding how multi-task data is introduced, how much multi-task data is introduced will affect the SuperGLUE downstream performance using the multi-task pretraining objective.

weakness and question:
- Given that SuperGLUE may represent one or several language understanding tasks, do the findings regards finetuning transfer, pre-finetuning hold still for other task families like generation? Or is that specifically tailored to SuperGLUE as mentioned in the paper.

- Section 2.5 uses the same learning rate for different batch sizes, which might result in unfair comparison given there are previous works suggesting how batch size, learning rate could have compound effects on the downstream tasks.

- In section 2.4, will R beyond 10 still lead to consistent improvement of the SuperGLUE performance is unclear.

- Beyond the empirical observations, are there explanations (like task affinity beyond the accuracy correlation) that could explain more clearly the results of co-training by multi-task finetuning on top of pre-trained models?

- The significance of the results on different task/task families can be further validated by reporting std/mean results with multiple seeds as in (Section 2.5)

- The zero-shot performance might also be a point to add given the multi-task prompting / finetuning seems to be helpful both for GPT-families and T5-families [1, 2].

[1] Wei, Jason, et al. "Finetuned language models are zero-shot learners." arXiv preprint arXiv:2109.01652 (2021).

[2] Sanh, Victor, et al. "Multitask Prompted Training Enables Zero-Shot Task Generalization." arXiv preprint arXiv:2110.08207 (2021).

**Summary Of The Paper:**

This paper revisits the idea of multi-task learning for natural language processing (NLP) and scales it up to 107 supervised NLP tasks as EXMIX (Extreme Mixture) across diverse domains and task families. It extensively analyzes the co-training effect between the different families of NLP tasks and proposes a model pre-trained using a multi-task objective of self-supervised span denoising and supervised EXMIX named EXT5. They show that a massive and diverse collection of pre-training tasks is generally preferable to an expensive search for the best combination of pre-training tasks. It further shows the effectiveness of EXT5 on a range of supervised NLP tasks.

**Summary Of The Review:**

The idea of leveraging multi-task learning for NLP has been introduced before as mentioned in the paper and not referred to as [3, 4]. It is great to revisit and extensively test this idea at scale at the paradigm of pre-training methods. It also offers potentials for future works on how existing labeled datasets can be used to further improve NLP models.

[3] McCann, Bryan, et al. "The natural language decathlon: Multitask learning as question answering." arXiv preprint arXiv:1806.08730 (2018).

[4] Subramanian, Sandeep, et al. "Learning general purpose distributed sentence representations via large scale multi-task learning." arXiv preprint arXiv:1804.00079 (2018).

---

> ### Author Response · Authors · 2021-11-16
> **Response to Reviewer 4oQF**
>
> We thank the reviewer for their feedback and suggestions. We attempt to address their concerns below:
>
> ### Missing citations
> We thank the reviewer for pointing us towards [McCann et al.](https://arxiv.org/abs/1806.08730) and [Subramanian et al.](https://openreview.net/forum?id=B18WgG-CZ), which we will cite in our revised manuscript (to be uploaded before the discussion period ends).
>
> ### Tail behavior of R
> We are running experiments using R=20 and plan on including them in our revised manuscript (before the discussion period ends).
>
> ### Task affinity beyond accuracy correlation
> While we believe that accuracy correlations are the most natural measure of empirical success, we also tried comparing gradient cosine similarities amongst different tasks’ validation sets for our ExT5 models, and found that to be a poor indicator of task affinity. [Vu et al.](https://arxiv.org/abs/2005.00770) explored using task embeddings to predict task affinity, but those are again only a proxy for empirical correlations.
>
> ### Mean/STD reporting for section 2.1
> We agree with the reviewer here, but it must also be noted that our task-family results involved 34 fine-tuning runs. While we would like to report results on 3-5 different seeds for each task-family pair, this would involve an additional 68 – 135 fine-tuning runs of 200k training steps each, which would be too computationally expensive.
>
> ### Tailoring learning rate to different batch sizes
> While we agree that the learning rate could have been tuned for smaller batch sizes, we note that the original T5 paper ([Raffel et al.](https://arxiv.org/abs/1910.10683)) also used the exact same hyperparameters for their analysis (see sec 3.1.2 of their paper), which resulted in strong performance. We also note that the results using a batch size of 128 are still strong (using 55 random tasks with B=128 matches using 0 tasks at B=512, which is unusual given that pre-training almost always benefits from the largest batch size possible).
>
> ### Zero shot performance
> Our focus for this work was on pre-trained representations, so we did not prioritise zero-shot evaluation. However, as a sanity check we ran some experiments on the dataset from [Hendrycks et al.](https://arxiv.org/abs/2009.03300), and our large and XL models (770M and 3B parameters) both got an average accuracy of around 38% (for reference, GPT-3 got 43.2% and 43.9% for 13B and 175B models respectively). We surmise that if zero-shot/prompting tasks are the goal, then an additional fine-tuning stage on top of our model might help, as T5's pre-training regimen is not naturally suited for prompting and might interfere unless it is "overwritten" by purely supervised fine-tuning.
>
> *Please let us know if our answers and clarifications addressed your concerns. We welcome any other comments or suggestions that will help us further improve the paper.*

---

> > ### Comment · Reviewer_4oQF · 2021-11-29
> > **thanks for the response**
> >
> > Thanks a lot for your response. I have read the feedback and other responses and decided to retain the original recommendation.

---

### Official Review · Reviewer_ri6r · 2021-11-03

**Correctness:** 3
**Technical Novelty And Significance:** 2
**Empirical Novelty And Significance:** 2
**Recommendation:** 5
**Confidence:** 4

**Main Review:**

Since the ideas presented are not new and were actually recently explored in both the T5 paper and Aghajanyan et al., (2021), I really view this paper as a scaling paper.

**Pros**

The paper presents interesting results on scaling T5 multi-task learning to close to 100 tasks, showing improvements on downstream benchmarks over vanilla T5.

**Cons**

- While the results improve performance over downstream benchmarks, I didn’t learn anything new from the paper. The paper completely reuses the T5 strategy for multi-task learning and just scales it up with more tasks. While this scaling could have been an interesting study, the ablations and empirical investigation don’t ask very interesting questions for this to be useful. I didn't find any significant recommendations for scaling tasks that can be reusable in future work or even applicable to models other than T5. Moreover, prior work (Aghajanyan et al., 2021) also studied most of these questions already, in the context of two pre-trained models, though in a slightly different setting. Over this and much other related work on multi-task and multi-lingual learning, the current paper only seems to contribute new results using the T5 model.
- Many of the results presented in motivating the work are over random subsets of pre-training tasks and random seeds for training. However, none of the results seem to give an idea of the variance of performance, nor any indication of how many different random subsets of tasks were used for the results.
- The task scaling experiment (2.5) seems to be core to the contribution of the paper. Similar scaling experiments are also present in Aghajanyan et al., (2021). However, apart from the lack of an exposition of the variance in the quantities presented, I found this analysis to be very underwhelming. The authors can analyze many interesting choices here to see how task scaling affects results. To give one instance, one can consider choices like selecting tasks proportional to their size (instead of random sampling) and evaluating how it affects performance. Without any meaningful consideration of the choices available here, I am not sure what future work can take away from this paper.
- Often the evaluation is in terms of “average” super-glue performance. Since many improvements are actually marginal (less than 1%), it will be good to provide a spread of the results (at least in the appendix) so the readers know the results are not due to improvement on just a small subset of downstream tasks. Many hyper-parameters and design choices for the pre-training are also difficult to find and not clearly specified.


**Questions and suggestions**

- Table 3: best effort has 48 tasks but random-mix is trained with 55, why? What is the variance of random-mix? What is the overlap between best-effort tasks and random tasks?
- 2.4: Is the amount of data and pre-training steps the same across the models being compared? If not, it will be good to make the amount of data and steps similar when comparing only C4 vs only C4 + ExMix. Moreover, it will actually be more useful to compare with respect to C4 performance at multiple data sizes, since pre-training data is more abundant than supervised multi-task data.
- Fig 3, adding multi-task over C4 only seems to help a little bit, looking at the spread of SuperGlue scores between the two settings will be more helpful. These numbers (at R=2) seem different than what is reported in Fig 5, can you comment on why?
- 2.5 and Fig 4: task scaling only helps at 512 batch-size, there doesn’t seem to be any consistent trend at batch-size of 128. This seems to indicate that batch-size is a very crucial axis for ablation. Did you try other batch-sizes, a plot of how performance varies with this parameter will be helpful. Moreover, batch-size here is confounded with the number of tasks that are sampled in a batch. What is the task sampling strategy used here and how does it affect this plot?
- Fig. 4: What is the shaded area? if it is some statistic like standard deviation across random seeds, why is this not present at 0 and 107.
- Fig. 4: What is the variance with respect to the random selection of tasks? Will a size-proportional selection of tasks lead to different results, for instance, to select 30 tasks pick the 30 with the largest data?
- In the super-glue results, improvements diminish at larger model sizes, to the extent that T5-XL performs similarly. This seems like an interesting result and the trend indicates multi-task might become less useful at even larger model sizes. It will be good to explore this further, if you have resources for such an experiment, as I consider this an important aspect of “scaling” which will fall within the scope of this work.
- Can you comment on your choice of T5 1.1 instead of the original T5 models published with the paper. The original models also include downstream supervised tasks in their pre-training, just like this work.

**Post Author Response**

I am not convinced that the experiments fully support the claim that adding a random mix of more tasks overcomes negative transfer between tasks. The spread of SuperGlue results with the number of tasks is highly inconsistent and performance of some tasks often gets worse at more tasks. The large variance in Fig 4 at 55 and 80 actually indicates task selection is really important to get higher performance, though the best-effort approach, as presented in this paper, might not be the best to get that task mixture. This then requires a more thorough evaluation of the various choices for multi-task task selection. I am also concerned with the large difference in trend at the two different batchsizes, which the authors didn't address. Compared to previous work, I have to disagree with the authors that problems like task heads or loss scaling are mitigated by T5. T5 still generates using a shared softmax classifier (over the whole vocabulary), instead confounding the problem with that of generation. Loss scaling is hidden in how you sample the instances/task for each batch. The main point of difference that I see from the rebuttal is of using multiple seeds in Fig 3. The response also included new results using a larger T5 model. The result on the larger T5 model seems to confirm the trend that there are diminishing returns from multi-tasking at a larger scale.

I am increasing my rating by one point (which in the new system makes it 5) as there are tidbits of interesting results in the paper, which if presented neutrally could have been interesting. However, I am concerned that this will be used as a point of reference to say that randomly adding more tasks always helps, for which the paper doesn't provide sufficient evidence.

**Summary Of The Paper:**

The paper presents an analysis of multi-task learning with the T5 model at a large scale. The tasks are formatted as text-to-text tasks and trained alongside the span denoising objective on C4, same as the multi-task strategy used in the original T5 paper. The paper evaluates this approach for a few different sizes of the T5 model on various downstream datasets, compared with vanilla T5 model. The paper shows improvements on average accuracy over multiple downstream benchmarks. In addition, there are some ablations on the number of tasks, the ratio of C4 vs multi-task, and comparing multi-task with pre-finetuning.

**Summary Of The Review:**

In general, I think it is important to carefully study multi-task scaling in the post pre-training era. There are many papers in a similar vein recently and to be useful the paper should concentrate more on asking meaningful questions and rigorously analyzing the results to provide recommendations for future work. I hope the authors take the criticism constructively to improve their work further, concentrating on more analysis rather than more downstream benchmark results.

---

> ### Author Response · Authors · 2021-11-16
> **Response to Reviewer ri6r (Part 1/2 -- Paper Novelty)**
>
> We thank the reviewer for their feedback and suggestions. We believe that there are three points that the reviewer might not have considered when evaluating this paper’s novelty:
>
> ### We do not believe that it is obvious that extreme task-scaling would work, especially for T5 models.
> In the original T5 paper ([Raffel et al.](https://arxiv.org/abs/1910.10683)), the authors found that including ~20 tasks in their multi-task pre-training setup did not significantly affect results (see section 3.5.3 in [Raffel et al.](https://arxiv.org/abs/1910.10683)). The conclusion of [Raffel et al.](https://arxiv.org/abs/1910.10683) (page 37) was “pre-training on a multi-task mixture of unsupervised and supervised tasks before fine-tuning worked as well as pre-training on the unsupervised task alone”. In fact, they later proposed T5.1.1, which does not use any supervised data while pre-training.
>
> This paper challenges that assertion, and shows that multi-task pre-training does indeed help when scaled. Moreover, [Aghajanyan et al.](https://arxiv.org/abs/2101.11038) had to modify the loss ([R3F/R4F](https://openreview.net/forum?id=OQ08SN70M1V)) for robust fine-tuning to get the model to perform well. [Aghajanyan et al.](https://arxiv.org/abs/2101.11038) and [Liu et al.]() also require task-specific heads and use more training steps than pre-trained models. Our model requires none of these, but rather shares all parameters across all tasks thanks to the seq2seq design. This allows us to directly measure the impact of multi-task scaling and have a lower risk of conflating it with other tweaks to model setup and more training steps.
>
>
> ### Many results and findings of this paper are new and can guide the development of future pre-trained large language models.
> In fact, our results contradict the findings of other recent studies that were performed in smaller scales such as T5 (see point above). We also compare to the pre-finetuning method of [Aghajanyan et al.](https://arxiv.org/abs/2101.11038) and show that extreme multi-task pre-training is better (section 2.3). Furthermore, our extensive analysis of different possible design choices, such as mixture ratio between supervised and unsupervised objectives, task choices, and batch size can guide future efforts. Finally, as far as we know, we are the first to show that multitasking improves sample efficiency for pre-training. While [Aghajanyan et al.](https://arxiv.org/abs/2101.11038) showed this for fine-tuning, pre-training is significantly more expensive. Therefore, we hope that our findings will help in reducing the significant cost of future pre-training efforts across the community ([Strubell et al., 2019](https://arxiv.org/abs/1906.02243)).
>
>
> ### The analyses present in this paper are significantly more substantial than prior works
> While [Aghajanyan et al.](https://arxiv.org/abs/2101.11038) and [Liu et al.](https://arxiv.org/abs/1901.11504) proposed competitive models using multi-task learning, we believe that our analyses are more conclusive. 1) Our task scaling experiments consider standard deviation over different random sets of tasks, which is absent in any prior works. 2) Our experiments explore how different mixing strategies for labeled data influence this process, which does not exist in prior works at this scale (e.g., pure supervised multi-task training (R=0) does not perform well, which emphasizes the importance of self-supervision). 3) Compared to prior work, our setup is simpler and doesn’t require modifying the loss function ([R3F/R4F](https://openreview.net/forum?id=OQ08SN70M1V) in [Aghajanyan et al.](https://arxiv.org/abs/2101.11038)) or task-specific heads/losses ([Aghajanyan et al.](https://arxiv.org/abs/2101.11038) and [Liu et al.](https://arxiv.org/abs/1901.11504)), which means our study isolates multi-task scaling better and has a lower risk of conflating it with other tweaks to model setup.

---

> ### Author Response · Authors · 2021-11-16
> **Response to Reviewer ri6r (Part 2/2 -- Specific Concerns)**
>
> We thank the reviewer for their feedback and suggestions. We attempt to address their specific concerns below:
>
> ### Task scaling experiment (2.5) compared to [Aghajanyan et al.](https://arxiv.org/abs/2101.11038)
> While [Aghajanyan et al.](https://arxiv.org/abs/2101.11038) also perform a task scaling experiment, there are three main differences that make our experiment interesting. First, and most importantly, the setting is different: [Aghajanyan et al.](https://arxiv.org/abs/2101.11038) perform scaling on pre-finetuning while we perform it on pre-training. Secondly, they report a single experiment per scale while we perform 3 runs with different random task subsamples and report variation in Figure 4. Finally, our task-scaling experiments are made more significant when paired with our findings in section 2.2 (our multi-task curation experiments).
>
> We agree that there are many interesting experiments that could be run but we leave them for future studies as we believe that our current experiments support our claims. Yet, we welcome and appreciate any suggestions. Regarding what can be learned for future work, we believe that our findings that simply adding more tasks at these scales is helpful, and is an important finding. We view the simplicity of randomly adding tasks rather than trying to pick a “best fit” of tasks for pre-training as an important advantage. This is what guided us to perform the expensive pre-training process only once (per architecture) with all tasks, and not to pre-train multiple “expert” pre-trained models with different task combinations. Also, as our out-of-mixture experiments show, our models benefit even new tasks. Therefore, as we state at the end of section 3, our main recommendation is to include as many available supervised tasks as possible, and to repeat the pre-training process periodically as more data becomes available.
>
> ### Best Effort Mixture vs Random Mixture
> The 48 tasks of the best-effort mixture are a result of combining all tasks from the four task families that had the highest results in the co-training experiments (Table 2). In contrast, the tasks for the random combinations (section 2.5) were picked randomly, without any prior knowledge on the relation between the pre-training task and the SuperGLUE tasks. The number 55 was simply chosen as approximately half the size of EXMIX for studying the effect of task scaling. We repeated each of the random tasks experiments (other than none and all) three times, sampling different tasks. The variance in the results is depicted in Figure 4.
>
> ### Is the amount of data and pre-training steps the same across the models compared?
> Yes, all of our comparisons keep the batch size and number of training steps consistent. This means that we include more labeled examples at the cost of pre-training examples, which enables a fairer comparison between models compared to prior work.
>
> ### Adding multi-tasking over C4 only seems to help a little bit?
> Yes, it must be noted that only using self-supervised C4 pre-training is a very strong baseline that was recently used in T5 to achieve state-of-the-art results on many tasks. As for the spread of SuperGLUE scores across individual datasets, we include the full SuperGLUE tables in our revised appendix (to be uploaded before the discussion period ends).
>
> ### Variance in FIgure 4, and random seeds
> The shaded area is the standard deviation across different random subsets. When selecting, say, 30 tasks out of 107, many different subsets are possible. So we report performance over three different random subsets for 30, 55, and 80 tasks resulting in 9 pre-training runs (per batch size). However, we acknowledge that the term “random seed” that we use to denote how different random subsets are selected can be confused with the random seed that is used to initialize the model variables, and will update our writing to be clearer.
> This is also why we can’t report standard deviation for 0 and 107 tasks, as the task selection is deterministic.
>
> ### Using T5.1.1 for pre-training instead of original T5 models
> The reason we opted for this is that using the original T5 models would not enable a comparison to pure self-supervised pre-training. As the reviewer notes, the original T5 models include a multi-task pre-training setup of around 20 tasks, so comparing with it in our analyses would not be a comparison to purely self-supervised pre-training.
>
> ### Higher model scale
> We will include additional SuperGLUE results on XXL-sized ExT5 models (about 11B parameters) in our revision (to be uploaded before the discussion period ends), which outperforms the equivalent T5.1.1 baselines.
>
> *Please let us know if our answers and clarifications addressed your concerns. We welcome any other comments or suggestions that will help us further improve the paper.*

---

### Official Review · Reviewer_u5L1 · 2021-11-03

**Correctness:** 4
**Technical Novelty And Significance:** 3
**Empirical Novelty And Significance:** 4
**Recommendation:** 8
**Confidence:** 4

**Main Review:**

strengths

- Well written and convincing
- The variety and size of their corpus ExMix
- The number of interesting research questions explored
- The improved efficiency of their approach as compared with the baseline methods
- They evaluate on a large number of varied text suites (SuperGLUE, GEM, Rainbow CommonSense, CBQA) and compare against a strong baseline T5 and mostly perform better.

weakness
- Nothing significant
- Selecting subsets of tasks and exploring fine-tuning (2.1, 2.2) takes up a lot of the paper and it did not perform as well as pre-training or impact the performance of pre-training much and so maybe less space could have been dedicated to these sections.



**Summary Of The Paper:**

This paper presents a corpus (ExMix) and a model (ExT5) which extend work on multi-task pretraining for large language models.

Their ExMix covers 8 families of supervised natural language processing tasks (eg. summarization, classification) with three individual tasks in each of them. These tasks are all converted into one text-to-text task (like prior work does) in order to allow uniform training on mutliple tasks. This is an impressively comprehensive collection of multi-task data sets, twice as large as the previously biggest multi-task corpus.

They also perform thorough experiments on a variety of intersting research questions.
They look at how to use cheaper fine-tuning to select a sub-set of tasks to use for pre-training - and they show that it is best to use all the supervised tasks in pretraining, not just the subset which worked best in fine-tuning.
They looked at whether adding and additional fine-tuning step before the final GLUE task fine-tuning would be better than pre-training with all this supervised multi-task data, and they show that in fact it is best to actually pre-train with them.
In their pre-training they mix masked-unsupervised learning with the task-supervised learning and they explore what is the right mix of the two, showing that most mixture settings are good, but that without the unsupervised signal, the model performs much worse. Also increasing the number of tasks generally helps downstream performance.

They also look at the efficiency of their approach. Training on ExMix allows the model to reach the performance of the T5 model trained only on plain text with many fewer (approximately half) the number of steps.

Finally they build a large pre-trained model ExT5 on which they perform really extensive evaluations and the demonstrate that the multi-task pretraining clearly helps and improves on a strong baseline across many tasks, including tasks which have not been included in the ExMix training data like translation.


**Summary Of The Review:**

This paper pushes forward work on large scale multi-task transfer learning, with a huge number and variety of tasks (107 for training) with many test suits and research questions explored. It is well written and convincing and I would like to see it published in ICLR.

---

> ### Author Response · Authors · 2021-11-16
> **Response to Reviewer u5L1**
>
> We thank the reviewer for their feedback and suggestions.
>
> The reason for allocating ~2 pages for sections 2.1 and 2.2 is that they motivate the design choices we made for our main experiments. We hope that our exploration, including the negative results, will be useful for any future development of pre-trained large language models.

---

### Author Response · Authors · 2021-11-19
**Summary of updates to our revised paper**

We thank the reviewers for their feedback. We summarize our revisions below:
* We include additional SuperGLUE results for our XXL-sized ExT5 model (around 11B parameters).
* We include experiments at the tail-end of the ratio of self-supervised C4 sampling rate to supervised ExMix sampling rate (R=20), which involved an additional pre-training + fine-tuning run (section 2.4).
* In Appendix C, we include the full SuperGLUE tables across individual datasets for our experiments in section 2.
* We include a few missing citations in the related works section (and we welcome any other suggestions for citations we might have missed).
* We update the writing/captions in section 2.5 to be clearer with respect to standard deviation reporting.
* We fix minor bugs in a few figures.

*Please let us know if our clarifications and revisions addressed your concerns. We welcome any other comments or suggestions that will help us further improve the paper.*

---

### Decision · Program_Chairs · 2022-01-20

**Decision:**

Accept (Poster)

**Comment:**

This paper explores large scale supervised multi-task training across 107 NLP tasks combined with self-supervised C4 masked span infilling, using the T5 sequence-to-sequence model.  The results improve over prior strong T5 baselines on several NLP benchmarks such as SuperGLUE, GEM, and Rainbow.

The paper's main strengths are the scale and large number of tasks, the release of the trained models and data, as well as the clarity and presentation.  Reviewers had concerns with the novelty, limitations in the evaluation (to just T5, and to just SuperGLUE in portions of the paper), and the potential impact of hyperparameters on the results.  During the discussion period, the authors noted that it is not obvious a priori that their approach would work, and that their evaluations on other tasks made it unlikely to be overfitting to SuperGLUE.  They also noted that running the additional hyperparameter experiments suggested during the reviews were computationally prohibitive.

Overall, despite the drawbacks and relative lack of novelty, the extensive experiments and released models provide significant value and will be of interest to the research community.